

# Associated bacteria of *Botryococcus braunii* (Chlorophyta)

Joao D. Gouveia[1,*], Jie Lian[2,*], Georg Steinert[2], Hauke Smidt[2], Detmer Sipkema[2], Rene H. Wijffels[1,3] and Maria J. Barbosa[1,4]

[1] Bioprocess Engineering, Wageningen University & Research, Wageningen, The Netherlands
[2] Laboratory of Microbiology, Wageningen University & Research, Wageningen, The Netherlands
[3] Faculty of Biosciences and Aquaculture, Nord University, Bodø, Norway
[4] Department of Biology, University of Bergen, Bergen, Norway
* These authors contributed equally to this work.

## ABSTRACT

*Botryococcus braunii* (Chlorophyta) is a green microalga known for producing hydrocarbons and exopolysaccharides (EPS). Improving the biomass productivity of *B. braunii* and hence, the productivity of the hydrocarbons and of the EPS, will make *B. braunii* more attractive for industries. Microalgae usually cohabit with bacteria which leads to the formation of species-specific communities with environmental and biological advantages. Bacteria have been found and identified with a few *B. braunii* strains, but little is known about the bacterial community across the different strains. A better knowledge of the bacterial community of *B. braunii* will help to optimize the biomass productivity, hydrocarbons, and EPS accumulation. To better understand the bacterial community diversity of *B. braunii,* we screened 12 strains from culture collections. Using 16S rRNA gene analysis by MiSeq we described the bacterial diversity across 12 *B. braunii* strains and identified possible shared communities. We found three bacterial families common to all strains: *Rhizobiaceae*, *Bradyrhizobiaceae*, and *Comamonadaceae*. Additionally, the results also suggest that each strain has its own specific bacteria that may be the result of long-term isolated culture.

## INTRODUCTION

In recent decades many studies have focused on the physiology and cultivation process of several microalgae with potential for large scale production (*Blanken et al., 2016*; *Cabanelas et al., 2016*; *Grima et al., 1999*; *Posten, 2009*; *Ugwu, Ogbonna & Tanaka, 2005*). One microalga of interest for large scale cultivation is *Botryococcus braunii* because it can produce and secrete long chain hydrocarbons and exopolysaccharides (EPS) (*Dayananda et al., 2007*; *Fernandes et al., 1989*; *Metzger & Largeau, 2005*). Hydrocarbons are naturally occurring compounds consisting entirely of hydrogen and carbon, and are one of the most important energy resources (*Timmis & Qin, 2010*) *B. braunii* is differentiated into different races (race A, B, L, and S) depending on the type of hydrocarbons secreted (*Kawachi et al., 2012*; *Metzger & Largeau, 2005*). Race A strains

Corresponding author
Joao D. Gouveia,
jdggouveia@gmail.com

synthesize odd-numbered alkadienes and trienes ($C_{25}$–$C_{31}$), race B strains synthesize isoprenoid type compounds termed botryococcenes ($C_{30}$–$C_{37}$), and methylated squalenes ($C_{31}$–$C_{34}$), race L strains synthesize lycopadiene ($C_{40}$), and race S strains synthesize $C_{18}$ epoxy-*n*-alkanes and $C_{20}$ saturated *n*-alkanes (*Dayananda et al., 2007*; *Eroglu, Okada & Melis, 2011*; *Kawachi et al., 2012*; *Metzger & Largeau, 2005*). EPS can have a range of applications, for example, it can be applied as stabilizers and gelling agents in food products. In addition, it has applications in the pharmaceutical and cosmeceutical industries (*Borowitzka, 2013*; *Buono et al., 2012*; *Donot et al., 2012*). *B. braunii* comprises of a variety of strains from diverse parts of the world. The strains can differ in the hydrocarbon and EPS content (*Allard & Casadevall, 1990*; *Dayananda et al., 2007*; *Eroglu, Okada & Melis, 2011*; *Gouveia et al., 2017*; *Metzger, Casadevall & Coute, 1988*; *Moutel et al., 2016*; *Volova et al., 1998*; *Wolf, 1983*).

Bacteria can grow in close proximity to the microalgal cells due to the presence of EPS substances secreted by the microalgae (*Bell & Mitchell, 1972*). The presence of bacteria within, or close to this EPS layer can lead to mutually beneficial interactions as well as interactions that are antagonistic in nature. Beneficial interactions for microalgae normally provide environmental advantages, such as nutrient exchange and community resilience to invasion by other species (*Eigemann et al., 2013*; *Hays et al., 2015*; *Jasti et al., 2005*; *Ramanan et al., 2015*). Antagonistic interactions will usually result in inhibition of the microalgal growth, either causing cell lysis, or directly competing for nutrients (*Cole, 1982*; *Cooper & Smith, 2015*; *Segev et al., 2016*). Studies investigating interactions of microalgae with bacteria show how important these interactions can be for the cultivation process (*Guerrini et al., 1998*; *Kazamia et al., 2012*; *Kim et al., 2014*; *Windler et al., 2014*). Understanding the interactions of microalgae and bacteria, and how it can enhance the cultivation for industrial process, could lead to increased biomass productivity.

So far, the bacterial community of *B. braunii* species is described in only a few studies. The earliest work is from *Chirac et al. (1982)* who described the presence of *Pseudomonas* sp. and *Flavobacterium* sp. in two strains of *B. braunii*. *Rivas, Vargas & Riquelme (2010)* identified in the *B. braunii* UTEX strain the presence of *Pseudomonas* sp. and *Rhizobium* sp. One study using the *B. braunii* Ba10 strain showed the presence of rod shaped bacteria in the rim of the colony aggregations and proposed it is as growth promoting bacteria closely related to *Hyphomonadaceae* spp. (*Tanabe et al., 2015*). One important finding was that *B. braunii* is a vitamin $B_{12}$ autotroph, so it does not depend on bacteria for the synthesis of this important metabolite (*Tanabe, Ioki & Watanabe, 2014*). A more recent study using a *B. braunii* (race B) strain, revealed the presence of several Rhizobiales such as *Bradyrhizobium*, and the presence of *Bacteroidetes* sp. (*Sambles et al., 2017*). So far, all studies have focused on only a few strains making it difficult to have a good overview of what bacterial community dominates *B. braunii*.

In this study, we looked at twelve strains of *B. braunii* obtained from several culture collections to investigate the bacterial community composition that is associated with *B. braunii*.

**Table 1 Information of the culture collections providers of *Botryococcus braunii* strains and location of origin.**

| Culture collection | *Botryococcus braunii* strain (our abbreviation) | Race | Location | Isolation, date of isolation | Reference |
|---|---|---|---|---|---|
| Berkeley | Showa | Race B | Culturing tanks, Berkley | By unknown, 1980 | *Nonomura (1988)* |
| Scandinavian Culture Collection of Algae and Protozoa (SCCAP) | K1489 | Race A | Belgium, Nieuwoort | By G. Hansen, 2008 | No reference |
| UTEX Culture Collection of Algae | UTEX LB572 (UTEX) | Race A | Cambridge, England | By M. R. Droop, 1950 | *Eroglu, Okada & Melis (2011)* |
| Culture Collection of Autotrophic Organisms (CCALA) | CCALA778 (CCALA) | Unknown | Serra da Estrela (Barragem da Erva da Fome) Portugal | By Santos, 1997 | No reference found |
| Culture Collection of Algae and Protozoa (CCAP) | CCAP807/2 (CCAP) | Race A | Grasmere, Cumbria, England | By Jaworski, 1984 | *Hilton, Rigg & Jaworski (1988)* |
| ALGOBANK-CAEN | AC755 | Race A | Lingoult-Morvan, France | By Pierre Metzger, 1981 | *Metzger et al. (1985)* |
| | AC759 | Race B | Ayame, Ivory Coast | By Pierre Metzger, 1984 | *Metzger, Casadevall & Coute (1988)* |
| | AC760 | Race B | Kossou, Ivory Coast | By Pierre Metzger, 1984 | *Metzger, Casadevall & Coute (1988)* |
| | AC761 | Race B | Paquemar, Martinique, France | By Pierre Metzger, 1983 | *Metzger et al. (1985)* |
| | AC765 | Race L | Kossou, Ivory Coast | By Pierre Metzger, 1984 | *Metzger, Casadevall & Coute (1988)* |
| | AC767 | Race L | Songkla Nakarin, Thailand | By Pierre Metzger, 1985 | *Metzger & Casadevall (1987)* |
| | AC768 | Race L | Yamoussoukro, Ivory Coast | By Pierre Metzger, 1984 | *Metzger & Casadevall (1987)* |

# EXPERIMENTAL PROCEDURE

## Strain collections and media preparation

Twelve *B. braunii* strains were obtained from culture collections (Table 1) and transferred to Erlenmeyer flasks with modified Chu 13 medium (*Largeau et al., 1980*) without citric acid or vitamins, with the following composition: 1,200 mg $L^{-1}$ $KNO_3$, 200 mg $L^{-1}$ $MgSO_4.2H_2O$, 108 mg $L^{-1}$ $CaCl_2.2H_2O$, 104.8 mg $L^{-1}$ $K_2HPO_4$, 20 mg $L^{-1}$ Fe-$Na_2$EDTA, 9.4 μg $L^{-1}$ $Na_2O_4Se$, 2.86 mg $L^{-1}$ $H_3BO_3$, 1.8 mg $L^{-1}$ $MnSO_4.4H_2O$, 220 μg $L^{-1}$ $ZnSO_4.7H_2O$, 90 μg $L^{-1}$ $CoSO_4.7H_2O$, 80 μg $L^{-1}$ $CuSO_4.5H_2O$, 60 μg $L^{-1}$ $Na_2MoO_4.2H_2O$, 10 μL $L^{-1}$ $H_2SO_4$. The final pH was adjusted to pH 7.2 with NaOH and $NaHCO_3$ was added to a final concentration of five mM. The 12 strains were grown in Infors HT Multitron incubators in 250 mL conical flasks and a volume of 150 mL. The temperature was set at 23 °C, with 2.5% $CO_2$ enriched air and shaking at 90 rpm. Illumination was provided by Phillips lamps FL-Tube L 36W/77, with 150 μmol photon $m^{-2}$ $s^{-1}$, and a light:dark photoperiod of 18:6 h. Flasks were inoculated with *B. braunii* growing in the

active growing phase, such that the initial absorbance at 680 nm was 0.2. The Erlenmeyer flasks were capped with aeraseal sterile film (Alphalabs, Hampshire, UK). Samples were taken at day one, four, eight, and 11, for 16S rRNA gene analyses.

## DNA extraction

On sampling days, five mL of fresh culture was harvested with sterilized membrane filters (0.2 µm; Merck-Millipore, Darmstadt, Germany) using a vacuum apparatus. The filters were cryopreserved in −80 °C until further processing. DNA was extracted from the cryopreserved filters that were cut into small pieces with a sterile scissor. Filter pieces were transferred to a two mL sterilized tube with zirconia/silica beads (Biospecs, Bartlesville, OK, USA), and one mL S.T.A.R buffer (Roche, Basel, Switzerland) was added. Cells were homogenized for two rounds of 45 s, at the speed of 5,500 rpm with Precellys (Bertin Technologies, Montigny le Bretonneux, France ). Then DNA was extracted using the Maxwell 16 Tissue LEV Total RNA purification kit (Promega, Madison, WI, USA) with aid of the Maxwell 16 instrument (Promega, Madison, WI, USA). The purity and quantity of DNA was examined by electrophoresis on a 1% agarose gel and measured with a Nanodrop (ND1000, Thermo Fisher Scientific Inc., Wilmington, Waltham, MA, USA). The extracted DNA was stored at −20 °C until further use.

## 16S rRNA gene amplification and Miseq sequencing

Amplicons from the V1–V2 region of 16S rRNA genes were generated by a two-step PCR strategy consisting of a forward primer (27F-DegS = 5′GTTYGATYMTGGCTCAG 3′ where M = A or C; R = A or G; W = A or T; Y = C or T) and an equimolar mixture of reverse primers (338R I = 5′GCWGCCTCCCGTAGGAGT 3′ and II = 5′ GCWGCC ACCCGTAGGTGT 3′ where M = A or C; R = A or G; W = A or T; Y = C or T). Eighteen bp Universal Tags 1 and 2 (Unitag1 = GAGCCGTAGCCAGTCTGC; Unitag2 = GCC GTGACCGTGACATCG) were appended at the 5′ end of the forward and reverse primer, respectively (*van den Bogert et al., 2011*; *Daims et al., 1999*; *Tian et al., 2016*). The first PCR mix (50 µL) contained 10 µL 5× HF buffer (Thermo Scientific™, Waltham, MA, USA), one µL dNTP Mix (10 mM; Promega, Leiden, the Netherlands), 1 U of Phusion® Hot Start II High-Fidelity DNA polymerase (Thermo Scientific™, Waltham, MA, USA), one µM of 27F-DegS forward primer, one µM of 338R I and II reverse primers, one µL template DNA and 32.5 µL nuclease free water. Amplification included an initial denaturation at 98 °C for 30 s; 25 cycles of denaturation at 98 °C for 10 s; annealing at 56 °C for 20 s and elongation at 72 °C for 20 s; and a final extension at 72 °C for 10 min. The PCR product size was examined by 1% gel electrophoresis. The second PCR mix (100 µL) contained 62 µL nuclease free water, five µL of PCR1 product, 20 µL 5× HF buffer, two µL dNTP Mix, 2 U of Phusion® Hot Start II High-Fidelity DNA polymerase, 500 nM of a forward and reverse primer equivalent to the Unitag1 and Unitag2 sequences, respectively, each appended with an eight nt sample specific barcode. Amplification included an initial denaturation at 98 °C for 30 s; five cycles of denaturation at 98 °C for 10 s, annealing at 52 °C for 20 s and elongation at 72 °C for 20 s; and a final extension at 72 °C for 10 min. The concentration of PCR products was quantified with a Qubit
Fluorometer (Life Technologies, Darmstadt, Germany) in combination with the dsDNA BR Assay kit (Invitrogen, Carlsbad, CA, USA). Purified products were then pooled in equimolar amounts of 100 ng μL$^{-1}$ and sequenced on a MiSeq platform (GATC-Biotech, Konstanz, Germany).

## Processing MiSeq data

Data was processed using the Quantitative Insights into Microbial Ecology 1.8.0. In short, paired-end libraries were filtered to contain only read pairs perfectly matching barcodes. Low quality or ambiguous reads were removed and then chimeric reads were removed and checked. Sequences with less than 0.1% were discarded. Remaining filtered sequences were assigned into operational taxonomy units (OTUs) at 97% threshold using an open reference method and a customized SILVA 16S rRNA gene reference (*Quast et al., 2013*). Seven samples from day 4 were removed from the results due to contamination during the PCR steps: AC755, AC759, AC760, AC767, AC768, CCAP, and UTEX572. The 16S rRNA gene dataset obtained in this study is deposited in the Sequence Read Archive, NCBI with accession number SRP102970.

## Microbial community analysis

For the interpretation of the microbial community data on family level, the OTU abundance table was converted to relative abundance and visualized as heatmaps using JColorGrid (*Joachimiak, Weisman & May, 2006*). Ordination analyses to estimate the relationship of the *B. braunii* strains based on dissimilarity of the microbial community compositions among the individual samples was performed for, (a) all strains of *B. braunii* used in this study, (b) all strains received from ALGOBANK-CAEN culture collection. For both analysis a standardized 97% OTU table (*decostand* function, *method = hellinger*) and the nMDS function metaMDS (*distance* = Bray–Curtis) from the vegan package in R was used (R version 3.0.2) (*Oksanen et al., 2016*; *R Core Team, 2014*). Beta dispersion and a permutation test were performed to test homogeneity dispersion within a group of samples. Adonis from the vegan package in R (v.3.0.2) was used to test significant differences in bacterial community between strains. Hierarchical clustering analysis was performed using hclust function in R using method = average.

## RESULTS

Figure 1 shows the bacterial families with a relative abundance above 1% and a total of four bacterial phyla associated with *B. braunii* strains. The four phyla found associated with *B. braunii* are the *Bacteroidetes, Gemmatimonadetes*, *Planctomycetes*, and *Proteobacteria*. *Proteobacteria* is the predominant bacterial phylum and representatives of this taxon are found in all 12 strains. *Bacteroidetes* is found in all strains with exception to strains AC761, AC768, and CCAP. *Gemmatimonadetes* is found only in the CAEN culture (with AC prefix) strains with exception to AC755. *Planctomycetes* is found in AC760, CCALA, K1489, Showa, and UTEX strains. Three families are found across all 12 *B. braunii* strains and all are *Proteobacteria*. These are the *Rhizobiaceae*, *Bradyrhizobiaceae*, and *Comamonadaceae*. *Rhizobiaceae* is represented by 1–59% of the

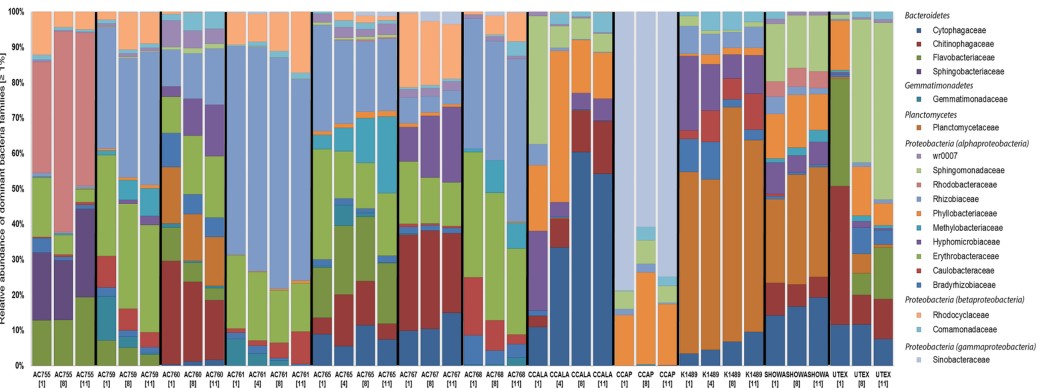

**Figure 1 Relative abundance of bacterial families in 12 *B. braunii* strains.** Strain abbreviations are used as explained in Table 1. Each bar displays the bacterial family relative abundance above 1%. Strains are labelled below with sample day within square brackets. Bacterial Families are organized according to the phyla (in italics) they belong to.               

bacterial reads. *Bradyrhizobiaceae* was found within the 1–8% range. *Comamonadaceae* was found between 1% and 5%. Two families of bacteria are only found in the strains obtained from the CAEN culture collection: *Erythrobacteraceae* with bacterial reads ranging from 1% to 29% and *Rhodocyclaceae* with 1–18%.

Some families of bacteria are particularly dominant in specific strains. *Sinobacteraceae* is dominant in CCAP with relative abundances ranging from 59% to 78%. *Planctomycetaceae* is dominant in K1489 strain with relative abundances between 46% and 51%. *Rhizobiaceae* is dominant in AC761 with relative abundances between 55% and 64%. Other families of bacteria become dominant as the cultures become older. *Rhodobacteraceae* is present in AC755 strain with relative abundances ranging from 28% at day 1 to 40% at day 11. *Sphingomonadaceae* is present in UTEX with 10% at day 1 and increases its presence to 47% at day 11. *Chytophagaceae* is dominant in CCALA strain with relative abundance ranging from 10% at day 1 to 52% at day 11.

Because we found three common families across all strains, we wanted to investigate in more detail the bacterial composition in these selected families and see if we could identify an unique microorganism present in all strains. Therefore, we zoomed in and looked at the OTUs distribution belonging to the three families: *Rhizobiaceae, Bradyrhizobiaceae*, and *Comamonadaceae*. In addition, we picked the OTUs found only in the strains obtained from the CAEN culture collection which belong to two families: *Erythrobacteraceae* and *Rhodocyclaceae*. The most abundant OTUs were selected and a total of 28 OTUs were investigated. From Fig. 2 it is clear that there is not an OTU that is found across all strains but rather each family comprises of several different OTUs. The second important observation is that CCAP strain has no representative OTUs for *Bradyrhizobiaceae* and *Rhizobiaceae* in the most abundant OTUs. The most represented family taxon is *Rhizobiaceae* with 12 OTUs. From the three families found in the 12 strains, OTU 233 assigned to the genus *Rhizobium* has the highest OTU frequency abundance with 10% and is present in seven out of 12 strains. The OTUs 143, 88, and 131 assigned to the genus *Shinella* are present in nine out of 12 strains. The OTUs 477, 475,

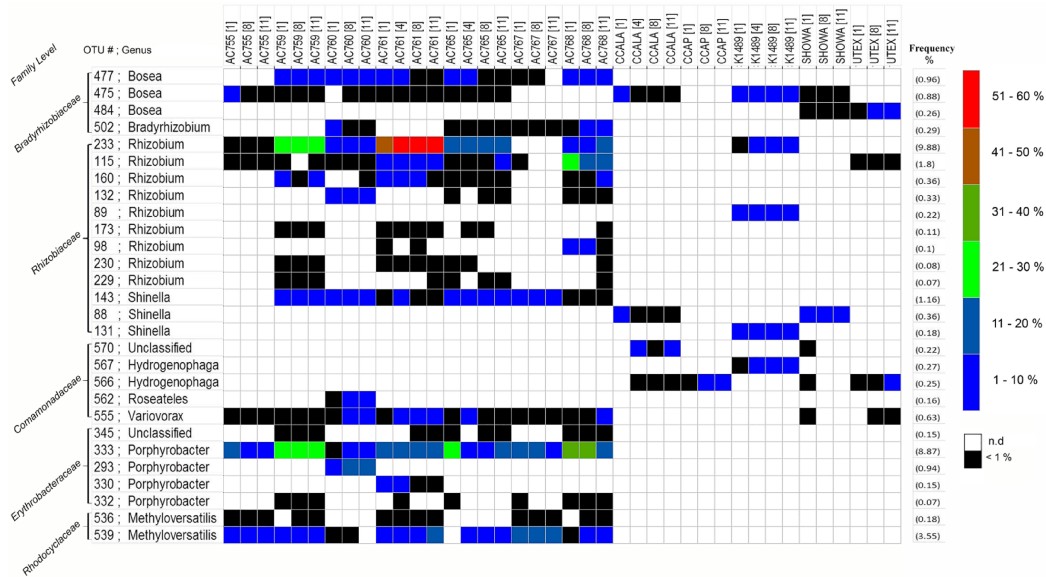

**Figure 2 Heatmap of most abundant 16S rRNA gene OTUs.** Label on the right show the color code for the relative abundance. Frequency (average relative abundance) of each OTU is shown in percentage on the right between brackets. Label on the left shows the family level and OTU number followed by genus. n.d, no reads detected.

and 484 assigned to the genus *Bosea* cover 11 out of 12 strains. From the two families found only in the cultures originating from the CAEN culture collection, OTUs 333 and 539 are found in all seven CAEN strains with an assigned genus *Porphyrobacter* and *Methyloversatilis*, respectively.

The most abundant OTUs (as listed in Fig. 2) were subjected to a Blast search against the NCBI database to infer their nearest neighbors (Table 2). OTUs 88, 115, 143, and 233 are similar in their nearest neighbors with four different *Rhizobium* spp. as candidates. Similar blast results are seen also for OTUs 566 and 567 with the nearest neighbors being *Hydrogenophaga* spp. The OTUs 819 and 832 with *Dyadobacter* spp. as nearest neighbor dominate CCALA bacterial community. Some OTUs show different species as closest neighbors such as OTUs 45 and 69 with *Frigidibacter albus*, *Paracoccus sediminis*, and *Nioella nitratireducens* as neighbors. The OTU 415 with high abundance in K1489 belonging to *Planctomycetaceae*, has as closest neighbors uncultured bacterium and third closest neighbor uncultured *Planctomyces* spp. with the latter showing 87% identity. The OTU 333 present only in the strains from CAEN culture collection, has 100% identity with *Sphingomonas* as closest two neighbors, and third neighbor, also with 100%, identity being *Porphyrobacter*.

Non-metric multidimensional scaling ordination was performed for the 12 strains to determine the bacterial community dissimilarities (Fig. 3A). *B. braunii* strains from the CAEN culture collection cluster together when compared to the other strains indicating these strains are similar to each other in bacterial community composition. This is supported by hierarchical cluster analysis showing CAEN strains in their own cluster (Fig. S1). The strains K1489, UTEX, CCAP, CCALA, and Showa represent separate clusters. The homogeneity of dispersion within each strain with 1,000 permutations

**Table 2  NCBI database blast of OTUs for selected families.**

| OTU | Nearest neighbor1 | Genbank acc. | Nearest neighbor2 | Genbank acc. | Nearest neighbor3 | Genbank acc. |
|---|---|---|---|---|---|---|
| 475 | *Hyphomicrobium nitrativorans* (100) | NR_121713.1 | *Hyphomicrobium nitrativorans* (100) | NR_118448.1 | *Bosea lathyri* (100) | NR_108515.1 |
| 477 | *Bradyrhizobium lupini* (100) | NR_134836.1 | *Bradyrhizobium lupini* (100) | NR_044869.2 | *Rhodopseudomonas palustris* (100) | NR_103926.1 |
| 484 | *Bosea robiniae* (100) | NR_108516.1 | *Bradyrhizobium lupini* (99) | NR_134836.1 | *Bradyrhizobium ottawaense* (99) | NR_133988.1 |
| 502 | *Bradyrhizobium daqingense* (100) | NR_118648.1 | *Bradyrhizobium lablabi* (100) | NR_117513.1 | *Beijerinckia doebereinerae* (100) | NR_116304.1 |
| 88 | *Rhizobium rhizoryzae* (100) | NR_133844.1 | *Rhizobium flavum* (100) | NR_133843.1 | *Rhizobium azibense* (100) | NR_133841.1 |
| 115 | *Rhizobium rhizoryzae* (100) | NR_133844.1 | *Rhizobium flavum* (100) | NR_133843.1 | *Rhizobium azibense* (100) | NR_133841.1 |
| 143 | *Rhizobium rhizoryzae* (100) | NR_133844.1 | *Rhizobium flavum* (100) | NR_133843.1 | *Rhizobium azibense* (100) | NR_133841.1 |
| 233 | *Rhizobium paranaense* (100) | NR_134152.1 | *Rhizobium rhizoryzae* (100) | NR_133844.1 | *Rhizobium flavum* (100) | NR_133843.1 |
| 555 | *Variovorax guangxiensis* (100) | NR_134828.1 | *Variovorax paradoxus* (100) | NR_074654.1 | *Variovorax boronicumulans* (100) | NR_114214.1 |
| 566 | *Hydrogenophaga flava* (100) | NR_114133.1 | Hydrogenophaga bisanensis (100) | NR_044268.1 | Hydrogenophaga defluvii (100) | NR_029024.1 |
| 567 | *Hydrogenophaga flava* (100) | NR_114133.1 | Hydrogenophaga bisanensis (100) | NR_044268.1 | Hydrogenophaga defluvii (100) | NR_029024.1 |
| 333 | *Sphingomonas gei* (100) | NR_134812.1 | *Sphingomonas ginsengisoli* (100) | NR_132664.1 | *Porphyrobacter colymbi* (100) | NR_114328.1 |
| 539 | Uncultured bacterium (100) | KY606782.1 | *Methyloversatilis discipulorum* (71) | KY284088.1 | *Methyloversatilis discipulorum* (71) | KY284080.1 |
| 63 | *Thioclava sp.* (100) | CP019437.1 | *Rhodobacter* sp. (100) | KY608089.1 | Uncultured *Rhodobacter* sp. (100) | KY606875.1 |
| 819 | *Dyadobacter jiangsuensis* (100) | NR_134721.1 | *Dyadobacter fermentans* (100) | NR_074368.1 | *Dyadobacter tibetensis* (88) | NR_109648.1 |
| 832 | *Dyadobacter jiangsuensis* (100) | NR_134721.1 | *Dyadobacter fermentans* (100) | NR_074368.1 | *Dyadobacter tibetensis* (88) | NR_109648.1 |
| 415 | Uncultured bacterium (100) | KT769749.1 | Uncultured bacterium (91) | KT724695.1 | Uncultured *Planctomyces* sp. (87) | JX576019.1 |
| 45 | *Frigidibacter albus* (100) | NR_134731.1 | *Paracoccus sediminis* (96) | NR_134122.1 | *Nioella nitratireducens* (94) | NR_134776.1 |
| 69 | *Frigidibacter albus* (100) | NR_134731.1 | *Paracoccus sediminis* (100) | NR_134122.1 | *Nioella nitratireducens* (97) | NR_134776.1 |
| 302 | *Sphingorhabdus arenilitoris* (100) | NR_134184.1 | *Sphingopyxis italica* (100) | NR_108877.1 | *Parasphingopyxis lamellibrachiae* (100) | NR_113006.1 |
| 310 | *Sphingomonas yantingensis* (100) | NR_133866.1 | *Sphingomonas canadensis* (100) | NR_108892.1 | *Blastomonas natatoria* (100) | NR_113794.1 |
| 355 | *Blastomonas natatoria* (100) | NR_113794.1 | *Sphingomonas ursincola* (100 | NR_040825.1 | *Blastomonas natatoria* (100) | NR_040824.1 |

**Note:**
Closest first three neighbors with highest identity match and with a minimum of 85% coverage for each OTU. NCBI blast on the February 11, 2016, except the OTU 662 which the blast search from August 30, 2016 and OTU 63 and 415 on February 2017.

show no significant difference ($F = 0.323$). Using adonis to test for bacterial community similarities between all strains, the results show that the bacterial communities are significantly different (DF = 11, Residuals = 28, $R^2 = 0.921$, $P = 0.001$). Figure 3B zooms in

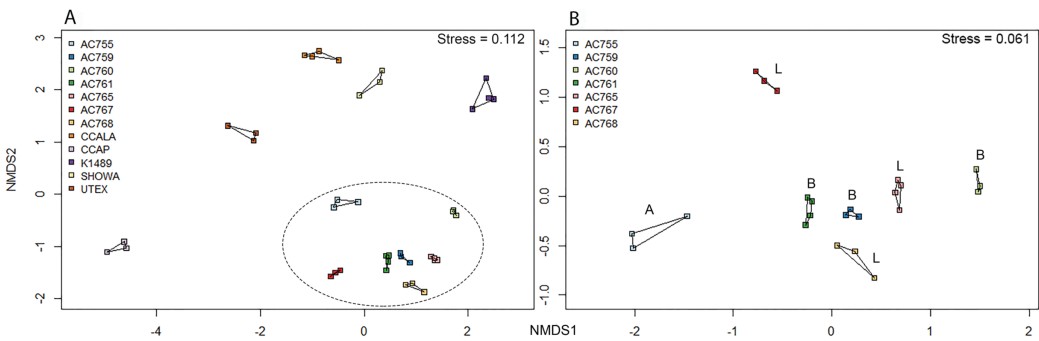

**Figure 3 Non-metric multidimensional scaling (nMDS) ordination (based on Bray–Curtis distance matrix) of 16S rRNA gene sequences of 12 *B. braunii* strains.** (A) Ordination of all strains with CAEN cultures clustering together (within the ellipse dotted line); (B) ordination of the CAEN culture collection strains only. Capital letters in plot (B) refer to the race subclassification based on the type of hydrocarbons produced.

to the CAEN culture collection strains. Races A, B, and L are subdivisions of *B. braunii* according to the type of hydrocarbons produced. No clustering by type of hydrocarbons produced was seen by the distribution of the race B and race L strains which are found mixed, namely race B AC759 and AC761 with race L AC765 and AC768. Similarly, the bacterial community between CAEN strains are significantly different (DF = 6, Residuals = 16, $R^2$ = 0.904, $P$ = 0.001).

## DISCUSSION

It is evident that *B. braunii* possesses a highly diverse bacterial community as seen by the range of bacterial phyla and families present in all the strains used in this study (Fig. 1; for a more comprehensive list see Fig. S2).

From the bacterial community analysis (Figs. 3A and 3B), it appears that each *B. braunii* strain has a specific bacterial community and no OTU is shared between all strains. The strains from the CAEN culture collection cluster together while *B. braunii* strains from other culture collections appear as separate groups. This implies that the culture collection from which the strain was obtained could potentially have an effect. With this study we are not able to really deduce the potential impact of the culture collection on the bacterial community because the experimental design was not set-up to do so. The presence of weak (within a culture collection) and strong (between culture collections) migration barriers may explain the bacterial profiles as obtained in our study and they may be a result of historical contingencies (*Fenchel, 2003*) rather than pointing toward highly specific interactions for a large number of OTUs. OTUs 539 and 333 are only found with the CAEN cultures and contributes toward these strains clustering in close proximity. OTU 333 is especially high in relative abundance and contributes to the distinctive clustering of the CAEN culture collection strains. The remaining strains also contain their specific OTUs that contribute toward their own clustering: OTU 819 and 832 with CCALA, OTU 310 with UTEX and K1489 with OTU 415. The bacterial community between three race B and three race L are mixed together (Fig. 3B). Therefore, no correlation was found between bacterial community and the type of hydrocarbons

produced between the two races. Similar observations were made in another study using six strains of *B. braunii* in which the authors did not find a correlation between the bacteria and type of hydrocarbon produced (*Chirac et al., 1985*).

Three bacterial families were found to be present with all twelve strains of *B. braunii*: *Bradyrhizobiaceae*, *Rhizobiaceae*, and *Comamonadaceae*. Two families were found abundantly only in the strains from the CAEN culture collection: *Erythrobacteraceae* and *Rhodocyclaceae*. The OTUs 88, 115, 143, and 233 blast hits show these are related to *Rhizobium* spp. (Table 2). *Rhizobium* spp. are known to form nodules in the roots of several plants within the family of legumes and are best known for nitrogen fixation. Nitrogen fixing bacteria were investigated in association with microalgae and it has been shown that they can enhance microalgae growth (*Hernandez et al., 2009*). *Rhizobium* spp. associated with *B. braunii* could have a similar role. *Rivas, Vargas & Riquelme (2010)* also found a *Rhizobium* sp. associated with *B. braunii* in particular UTEX LB572, and *Kim et al. (2014)* showed the presence of *Rhizobium* sp. with *B. braunii* 572. *Sambles et al. (2017)* identified *Rhizobium* sp. closely associated with *B. braunii* after submitting the cultures through a wash step and antibiotic treatment. Recent studies also shows *Rhizobium* spp. present with *Chlamydomonas reinhardtii*, *Chlorella vulgaris*, and *Scenedesmus* spp. (*Kim et al., 2014*). *Rhizobium* spp. seem important to *B. braunii* strains as it appears in all 12 strains with more prominence in the CAEN cultures and K1489 with three to four OTUs (Fig. 2). For the remaining strains CCALA, CCAP, Showa, and UTEX, *Rhizobium* spp. is represented only with one OTU.

Operational taxonomy unit 475 from *Bradyrhizobiaceae* family shows 100% similarity with the species *Hyphomicrobium nitrativorans* as the two closest neighbors and is present in 10 out of 12 *B. braunii* strains. *H. nitrativorans* is a known denitrifier isolated from a seawater treatment facility (*Martineau et al., 2013*). Denitrification is the process of reducing nitrate into a variety of gaseous compounds with the final being dinitrogen. Because denitrification mainly occurs in the absence of oxygen it is unlikely that this is happening within our cultures that are well oxygenated. The third closest neighbor for OTU 475 is *Bosea lathyri* and is associated with root nodules legumes (*De Meyer & Willems, 2012*).

Operational taxonomy units 555, 566, and 567 from *Comamonadaceae* family, appeared in seven out of 12 strains. The three closest neighbors of OTU 555 were *Variovorax* spp. and for OTUs 566 and 567 these were *Hydrogenophaga* spp., *Variovorax*, and *Hydrogenophaga* spp. are not known for being symbionts but may be able to support ecosystems by their ability to degrade toxic compounds and assist in nutrient recycling, therefore potentially producing benefits to other microorganisms (*Satola, Wübbeler & Steinbüchel, 2013*; *Yoon et al., 2008*). *Comamonadaceae* also appeared as one of the main bacteria families associated with cultivation of microalgae in bioreactors using a mix of fresh water and municipal water as part of a water treatment strategy (*Krustok et al., 2015*).

*Erythrobacteraceae* and *Rhodocyclaceae* were only found in the strains from CAEN culture collection. OTU 333 (*Erythrobacteraceae*) first two closest neighbors are from *Sphingomonas* spp., and third closest neighbor is *Porphyrobacter* spp. isolated from water in a swimming pool. Most *Porphyrobacter* spp. isolated originate from aquatic

environments (*Tonon, Moreira & Thompson, 2014*) and are associated with fresh water sediments (*Fang et al., 2015*). *Porphyrobacter* spp. have also been associated with other microalgae such as *Tetraselmis suecica* (*Biondi et al., 2016*). OTU 539 (*Rhodocyclaceae*) second and third closest neighbor is *Methyloversatilis discipulorum* which is a bacteria found in biofilms formation in engineered freshwater installations (*Van Der Kooij et al., 2017*). It is not clear why OTU 333 and 539 are specifically found only in the strains originating from the CAEN culture collection, but it could be an introduced species during handling. None the less, these two OTUs are present in high relative abundance (Fig. 2), and would be interesting to know if they have a positive or negative influence on the growth of the CAEN strains. It would be interesting to confirm such statement by attempting the removal of these OTUs and investigate the biomass growth.

*Sinobacteraceae* is dominant in CCAP (Fig. 1). This family was proposed in 2008 with the characterization of a bacteria from a polluted soil in Chi (*Zhou et al., 2008*). A recent bacteria related to hydrocarbon degradation shows similarities with *Sinobacteraceae* (*Gutierrez et al., 2013*). OTU 63 is highly abundant in CCAP and could have a negative impact in the cultivation of CCAP strain by reducing its hydrocarbon content.

The *Bactoroidetes* family *Cytophagaceae* dominates the culture CCALA at later stages of growth (Fig. 1). *Cytophagaceae* has also been found present in laboratory scale photobioreactor cultivation using wastewater for production of microalgae biomass (*Krustok et al., 2015*). The two OTUs that dominate the bacterial community in CCALA are OTU 819 and OTU 832. The Blast search on NCBI database approximates these two OTUs as *Dyadobacter* spp. which have also been found co-habiting with *Chlorella* spp. (*Otsuka et al., 2008*).

*Planctomycetaceae* dominates the bacterial community in K1489 strain (Fig. 1) with one OTU 415. This family can be found in freshwater biofilms and also strongly associated with macroalga (*Abed et al., 2014*; *Lage & Bondoso, 2014*). Species in this family could possibly be involved in metallic-oxide formation and be co-players in sulphate-reduction with the latter also involving a sulfur-reducing bacteria (*Shu et al., 2011*).

*Rhodobacteraceae* is present with up to 55% of bacterial relative abundance in AC755. Members of this family have been also isolated from other microalgae, namely *Chlorella pyrenoidosa* and *Scenedesmus obliquus* (*Schwenk, Nohynek & Rischer, 2014*). The OTUs 45 and 69 blast searches in NCBI database show the closest neighbors to be *F. albus*, *P. sediminis*, and *N. nitratireducens* (Table 2). All three neighbors were isolated from water environments (*Li & Zhou, 2015*; *Pan et al., 2014*).

*Sphingomonadaceae* is mostly found in freshwater and marine sediments (*Newton et al., 2011*). OTUs 302, 310, and 355 from this family were found in 6 out of 12 strains above 1% relative abundance. OTU 310 is only found in the UTEX strain with *Sphingomonas* spp. as the two closest neighbors. *Sphingomonas* spp. are shown to co-habit with other microalgae such as *Chlorella sorokiniana* and *Chlorella vulgaris* (*Ramanan et al., 2015*; *Watanabe et al., 2005*). *Sphingomonas* spp. have been shown to be able to degrade polycyclic aromatic hydrocarbons (*Tang et al., 2010*) and could possibly be degrading the hydrocarbons secreted by *B. braunii* as its carbon source.

Another characteristic of many bacteria is the ability to produce EPS such as species from the *Rhizobiaceae* and *Bradyrhizobiaceae* family (*Alves, De Souza & Varani, 2014*; *Bomfeti et al., 2011*; *Freitas, Alves & Reis, 2011*). This characteristic could play a role on the colony aggregation of *B. braunii* as EPS is known to be essential for biofilm formation (*Flemming, Neu & Wozniak, 2007*). Therefore, it would be interesting in the future to study this possible relationship as *B. braunii* is a colony forming organism. Such studies could involve the introduction of bacteria associated with colony formation such as *Terramonas ferruginea* as it has been associated with inducing flocculation in *Chlorella vulgaris* cultures (*Lee et al., 2013*).

With the present high microbial diversity, *B. braunii* shows qualities in resilience toward microbial activity, probably due to its colonial morphology and protective phycosphere made of hydrocarbons and EPS (*Weiss et al., 2012*). A number of microbes are potentially beneficial such as *Rhizobium* spp. which have been shown to have a positive effect on the biomass productivities of *B. braunii* UTEX (*Rivas, Vargas & Riquelme, 2010*), and *Hydrogenophaga* with the ability to degrade toxic compounds (*Yoon et al., 2008*). There are also microbes that may cause detrimental effects on hydrocarbon productivities of *B. braunii* such as *Sphingomonas* spp. (OTU 310) with its ability to degrade hydrocarbons (*Tang et al., 2010*). The removal of such detrimental microbes could enhance cultivation allowing more nitrogen available for biomass production and increase hydrocarbon accumulation of *B. braunii* as well as EPS production at a larger industrial scale.

## CONCLUSION

*Botryococcus braunii* can host a diverse microbial community and it is likely that some form of interaction is taking place with the members from the *Rhizobiaceae*, *Bradyrhizobiaceae*, and *Comamonadaceae* family, which all belong to the phylum *Proteobacteria*. There is not a specific bacterial community correlated to the different types of hydrocarbons produced by race B and L and mostly likely also not race A. *B. braunii* has many strains and each seems to have its own species-specific bacterial community. With a diverse microbial community present, it is also likely that some bacteria are having antagonistic effects on *B. braunii* such as competition with nutrients and degradation of hydrocarbons. *Botryococcus* is a microalgae of high scientific interest and it is important to understand better the associated bacteria. *Botryococcus*-associated bacteria are hard to get rid of (J. Gouveia, 2016, unpublished data) and therefore, it is important to start mass cultivation without those bacteria that are most harmful to the process.

### Funding

This project is carried out with financial support from the European Community under the seventh framework programme (Project SPLASH, contract nr. 311956), and Jie Lian was supported by the China Scholarship Council (No. 201406310023). The funders

had no role in study design, data collection and analysis, decision to publish, or preparation of the manuscript.

### Grant Disclosures

The following grant information was disclosed by the authors:
European Community under the seventh framework programme: Project SPLASH, contract nr. 311956.
China Scholarship Council: 201406310023.

### Competing Interests

Hauke Smidt is an Academic Editor for PeerJ.

### Author Contributions

- Joao D. Gouveia conceived and designed the experiments, performed the experiments, analyzed the data, contributed reagents/materials/analysis tools, prepared figures and/or tables, authored or reviewed drafts of the paper, approved the final draft.
- Jie Lian conceived and designed the experiments, performed the experiments, analyzed the data, contributed reagents/materials/analysis tools, prepared figures and/or tables, authored or reviewed drafts of the paper, approved the final draft.
- Georg Steinert conceived and designed the experiments, analyzed the data, contributed reagents/materials/analysis tools, prepared figures and/or tables, authored or reviewed drafts of the paper, approved the final draft.
- Hauke Smidt conceived and designed the experiments, analyzed the data, prepared figures and/or tables, authored or reviewed drafts of the paper, approved the final draft.
- Detmer Sipkema conceived and designed the experiments, analyzed the data, prepared figures and/or tables, authored or reviewed drafts of the paper, approved the final draft.
- Rene H. Wijffels conceived and designed the experiments, analyzed the data, prepared figures and/or tables, authored or reviewed drafts of the paper, approved the final draft.
- Maria J. Barbosa conceived and designed the experiments, analyzed the data, prepared figures and/or tables, authored or reviewed drafts of the paper, approved the final draft.

### Data Availability

The 16S rRNA gene dataset obtained in this study is available in the Sequence Read Archive, accession number SRP102970.

### Supplemental Information

Supplemental information for this article can be found online at http://dx.doi.org/10.7717/peerj.6610#supplemental-information.

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
