# Peer review of "Associated bacteria of Botryococcus braunii (Chlorophyta)"

_PeerJ, doi:10.7717/peerj.6610_

## Round 0.1 · original submission · Minor Revisions

Please provide a detailed point-by-point reply to all of the reviewers' comments along with your revised manuscript.

Reviewer 1 ·

Basic reporting

Excellent level and good clarity.
Suitable references.

Experimental design

Yes. All good.

Validity of the findings

Conclusions are not all supported by the evidence and work.
Speculative comments not identified as such.

Additional comments

I have read with interest the manuscript by Gouveia et al., entitled “Associated Bacteria of Botryococcus braunii (Chlorophyta).
The paper describes a 16SrDNA barcode approach to describing the bacterial consortium that is present in different, long-term cultures of B. braunii. Essentially, this study seems to follow on from previous work published by Sambles et al. (2017) in Microbiology Open (6; e482) on which Gouveia is an author but does not refer to this in any way. This lack is surprising given that the earlier investigation demonstrated the adequacy of the 16S rDNA approach relative to whole-genome, shortgun sequencing, and resulted in an almost identical conclusion; that B. braunii appears closely associated with members of the rhizobiales that are possibly involved in nitrogen assimilation.
Another important observation in this paper is that there are more similarities in bacterial demographies within culture collections than between any B. braunii phylogeny – the exception being the aforementioned rhizobiales (lines 237-239). This observation convinces me that the bacteria present is each culture are more probably due to culture artefacts than to any, normal microconsortium. These artefacts are interesting, but well known in algal culture, and therefore the paper really does not present any new insight into algal biology on microbial associations. Due to these artefacts, it is also impossible to assert, as the authors do, that “each strain has its own specific bacteria that are not shared between strains [of Botryococcus]” (lines 34-35); rather, it is clear that long-term, isolated culture leads to divergent representation in restricted populations that is dependent on the founder species – again a well-known tenet of ecology.
The paper, therefore, is underwhelming in its scope and conclusions, offering little new beyond the demographic description of bacteria present at a specific moment in long-term Botrycocccus cultures. The science, however, is correct and well described, with no major flaws. Equally, the manuscript is well-written and the figures are clear and representative of the data.
I cannot reject the paper on the basis of the experimentation or the methods, but I do not believe that it is, in of itself, worthy of publication, as the insights offered are much too slender and not generally applicable beyond the precise cultures used.

Reviewer 2 ·

Basic reporting

no comment

Experimental design

no comment

Validity of the findings

no comment

Additional comments

The manuscript is scientifically interesting and I recommend it for publication after minor revisions.
Specific comments
Line 62: “… (Jasti et al. 2005, Ramanan et al. 2015) (Eigemann et al. 2013, Hays et al. 2015)”. Is there any reason why the above citations are given in different parentheses?
line 78: “…B. braunii sp.” Delete the abbreviations ‘sp.’
line 204: “..Hydrogenophaga spp..”. Delete the second dot after spp.
line 274: “..were Hydrogenophaga spp.. Variovorax..” Replace the second dot with a comma.
Line 285: “..such as Chlorella spp. (Biondi et al. 2016).” Please correct, this citation refers to Tetraselmis not to Chlorella.
Lines 319-320. “ .. such as C. sorokiniana..” Please spell out the name of genus in full.

Lines 345-347. The authors claim that “There is not a specific bacterial community
correlated to the different types of hydrocarbons produced by race A and B and mostly likely also not race L and S.” Why the authors refer to race S? This statement cannot be concluded with the analysis and results of the article.

Figure 3. I suggest to the authors to add the race subclassification letter symbols on plot (a) as they did in plot (b).

Table 1. Please pay attention to: a) Citations not included in the reference list (e.g. Nonomura 1988, Metzger et al 1985, Hilton et al 1989, Metzger and Casadevall 1987). b) Formatting of citations.

---

## Round 0.2 · accepted · Accept

Thank you for your revision. Looking forward to receiving your future works.

# Reviewer 2 ·

Basic reporting

Clear and unambiguous

Experimental design

Methods described with sufficient detail.

Validity of the findings

Data is robust. Conclusions are well stated linked to original research question.

Additional comments

The revision on the paper and the responses to my comments are satisfactory.